# Neural-Symbolic Recursive Machine for Systematic Generalization

**Qing Li**[1], **Yixin Zhu**[3], **Yitao Liang**[1,3], **Ying Nian Wu**[2], **Song-Chun Zhu**[1,3], **Siyuan Huang**[1]
[1]National Key Laboratory of General Artificial Intelligence, BIGAI
[2]Department of Statistics, UCLA
[3]Institute for Artificial Intelligence, Peking University
https://liqing-ustc.github.io/NSR

## Abstract

Current learning models often struggle with human-like systematic generalization, particularly in learning compositional rules from limited data and extrapolating them to novel combinations. We introduce the Neural-Symbolic Recursive Machine (NSR), whose core is a Grounded Symbol System (GSS), allowing for the emergence of combinatorial syntax and semantics directly from training data. The NSR employs a modular design that integrates neural perception, syntactic parsing, and semantic reasoning. These components are synergistically trained through a novel deduction-abduction algorithm. Our findings demonstrate that NSR's design, imbued with the inductive biases of *equivariance* and *compositionality*, grants it the expressiveness to adeptly handle diverse sequence-to-sequence tasks and achieve unparalleled systematic generalization. We evaluate NSR's efficacy across four challenging benchmarks designed to probe systematic generalization capabilities: SCAN for semantic parsing, PCFG for string manipulation, HINT for arithmetic reasoning, and a compositional machine translation task. The results affirm NSR's superiority over contemporary neural and hybrid models in terms of generalization and transferability.

## 1 Introduction

A defining characteristic of human intelligence, *systematic compositionality* (Lake and Baroni, 2023), represents the algebraic capability to generate infinite interpretations from finite, known components, famously described as the "infinite use of finite means" (Chomsky, 1957; Montague, 1970; Marcus, 2018; Xie et al., 2021). This principle is essential for extrapolating from limited data to novel combinations (Lake et al., 2017; Jiang et al., 2023). To evaluate machine learning models' ability for systematic generalization, datasets such as SCAN (Lake and Baroni, 2018), PCFG (Hupkes et al., 2020), CFQ (Keysers et al., 2020), and Hint (Li et al., 2023b) have been introduced. Traditional neural networks often falter on these challenges, leading to the exploration of inductive biases to foster better generalization. Innovations like relative positional encoding and layer weight sharing have been proposed to improve Transformers' generalization capabilities (Csordás et al., 2021; Ontanón et al., 2022). Moreover, neural-symbolic stack machines have been demonstrated to achieve remarkable accuracy on tasks similar to SCAN (Chen et al., 2020), and large language models have been guided towards compositional semantic parsing on CFQ (Drozdov et al., 2023). Yet, these solutions usually require domain-specific knowledge and struggle with domain transfer.

In pursuit of achieving human-like systematic generalization across various domains, we introduce the Neural-Symbolic Recursive Machine (NSR), a principled framework designed for the *joint* learning of perception, syntax, and semantics. At the heart of NSR lies a Grounded Symbol System (GSS), depicted in Fig. 1, which emerges solely from the training data, eliminating the need for domain-specific knowledge. NSR employs a modular design, integrating neural perception, syntactic parsing, and semantic reasoning. Initially, a neural network acts as the perception module, grounding symbols in raw inputs. These symbols are then organized into a syntax tree by a transition-based neural dependency parser (Chen and Manning, 2014), followed by the interpretation of their semantic meaning using functional programs (Ellis et al., 2021). We prove that NSR's capacity for various sequence-to-sequence tasks, underpinned by the inductive biases of equivariance and compositionality, allows for the decomposition of complex inputs, sequential processing of components, and their

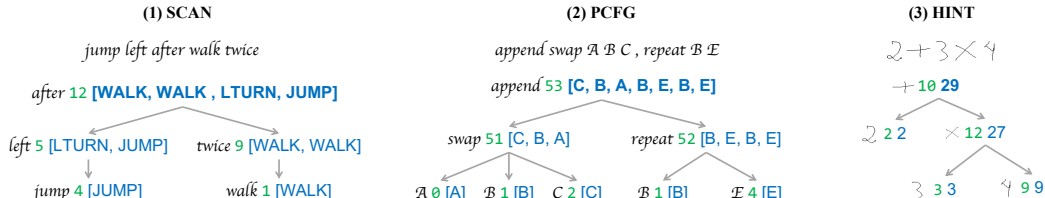

Figure 1: **Illustrative examples of GSSs demonstrating human-like reasoning processes.** (a) SCAN: each node encapsulates a triplet (*word*, symbol, action sequence). (b) PCFG: nodes consist of triplets (*word*, symbol, letter list). (c) HINT: nodes contain triplets (*image*, symbol, value). Symbols are denoted by their indices.

recomposition, thus facilitating the acquisition of meaningful symbols and compositional rules. These biases are critical for NSR's exceptional systematic generalization.

The end-to-end optimization of NSR poses significant challenges due to the lack of annotations for the internal GSS and the model's non-differentiable components. To circumvent these obstacles, we devise a probabilistic learning framework and a novel *deduction-abduction* algorithm for the joint training. This algorithm starts with a greedy deduction to form an initial GSS, which might be inaccurate. A top-down search-based abduction follows, aiming to refine the initial GSS by exploring its neighborhood for better solutions until the correct result is obtained. This refined GSS acts as *pseudo* supervision, enabling the independent training of each NSR component.

NSR's performance is validated against three benchmarks that test systematic generalization:

1. SCAN (Lake and Baroni, 2018), for translating natural language commands to action sequences;
2. PCFG (Hupkes et al., 2020), for predicting output sequences from string manipulation commands;
3. HINT (Li et al., 2023b), for computing results of handwritten arithmetic expressions.

NSR establishes new records on these benchmarks, achieving 100% generalization accuracy on SCAN and PCFG, and surpassing the previous best accuracy on HINT by 23%. Our analyses show that NSR's modular design and intrinsic inductive biases lead to stronger generalization than traditional neural networks and enhanced transferability over existing neural-symbolic models, with reduced need for domain-specific knowledge. NSR's effectiveness is further demonstrated on a compositional machine translation task by Lake and Baroni (2018) with a 100% generalization accuracy, revealing its potential in practical applications with complex and ambiguous rules.

## 2 RELATED WORK

The exploration of systematic generalization within deep neural networks has captivated the machine learning community, especially following the introduction of the SCAN dataset (Lake and Baroni, 2018). Various benchmarks have since been introduced across multiple domains, including semantic parsing (Keysers et al., 2020; Kim and Linzen, 2020), string manipulation (Hupkes et al., 2020), visual question answering (Bahdanau et al., 2019; Xie et al., 2021), grounded language understanding (Ruis et al., 2020), mathematical reasoning (Saxton et al., 2018; Li et al., 2023b), physical reasoning (Li et al., 2024a; Dai et al., 2023; Li et al., 2022), word learning (Jiang et al., 2023), robot manipulation (Li et al., 2024b; Wang et al., 2024; Li et al., 2023a), and recently developed open-ended worlds (Fan et al., 2022; Xu et al., 2023), serving as platforms to evaluate different aspects of generalization, like systematicity and productivity. In particular, research in semantic parsing (Chen et al., 2020; Herzig and Berant, 2021; Drozdov et al., 2023) has explored incorporating diverse inductive biases into neural networks to improve performance across these datasets. We categorize these approaches into three groups based on their method of embedding inductive bias:

1. **Architectural Prior:** Techniques under this category aim to refine neural architectures to foster compositional generalization. Dessì and Baroni (2019) have shown convolutional networks' effectiveness over RNNs in SCAN's "jump" split. Russin et al. (2019) improve RNNs by developing separate modules for syntax and semantics, while Gordon et al. (2019) introduce a permutation equivariant seq2seq model with convolutional operations for local equivariance. Enhancements in Transformers' systematic generalization through relative positional encoding and layer weight sharing are reported by Csordás et al. (2021) and Ontanón et al. (2022). Furthermore, Gontier et al. (2022) embed entity type abstractions in pretrained Transformers to boost logical reasoning.

2. **Data Augmentation:** This strategy devises schemes to create auxiliary training data to foster compositional generalization. Andreas (2020) augment data by interchanging fragments of training samples, and Akyürek et al. (2020) employ a generative model to recombine and resample training data. The meta sequence-to-sequence model (Lake, 2019) and the rule synthesizer (Nye et al., 2020) utilize samples from a meta-grammar resembling the SCAN grammar.
3. **Symbolic Scaffolding:** This strategy entails the integration of symbolic components into neural frameworks to enhance generalization. Liu et al. (2020) link a memory-augmented model with analytical expressions, emulating reasoning processes. Chen et al. (2020) embed a symbolic stack machine within a seq2seq model, with a neural controller managing operations. Kim (2021) derive latent neural grammars for both the encoder and decoder in a seq2seq model. While these methods achieve significant generalization by embedding symbolic scaffolding, their reliance on domain-specific knowledge and complex training methods, such as hierarchical reinforcement learning in Liu et al. (2020) and exhaustive search in Chen et al. (2020), restrict their practical use.

Beyond these technical implementations, existing research underscores two core inductive biases critical for compositional generalization: *equivariance* and *compositionality*. The introduced NSR model incorporates a generalized Grounded Symbol System as symbolic scaffolding and instills these biases within its modules, facilitating robust compositional generalization. Distinct from prior neural-symbolic methods, NSR requires minimal domain-specific knowledge and forgoes a specialized learning curriculum, leading to enhanced transferability and streamlined optimization across different domains, as demonstrated by our experiments; we refer readers to Sec. 4 for details.

Our work also intersects with neural-symbolic approaches for logical reasoning, such as the introduction of Neural Theorem Provers (NTPs) by Rocktäschel and Riedel (2017) for end-to-end differentiable proving with dense vector representations of symbols. Minervini et al. (2020) address NTPs' computational challenges with Greedy NTPs, extending their application to real-world datasets. Furthermore, Mao et al. (2018) present a Neuro-Symbolic Concept Learner that utilizes a domain-specific language to learn visual concepts from question-answering pairs.

## 3 NEURAL-SYMBOLIC RECURSIVE MACHINE

### 3.1 REPRESENTATION: GROUNDED SYMBOL SYSTEM (GSS)

The debate between connectionism and symbolism on the optimal representation for systematic generalization has been longstanding (Fodor et al., 1988; Fodor and Lepore, 2002; Marcus, 2018). Connectionism argues for *distributed representations*, where concepts are encoded across many neurons (Hinton, 1984), while symbolism advocates for *physical symbol systems*, with each symbol encapsulating an atomic concept, and complex ideas constructed through syntactical combination (Newell, 1980; Chomsky, 1965; Hauser et al., 2002; Evans and Levinson, 2009). Symbol systems, due to their interpretability, enable higher abstraction and generalization capabilities than distributed representations (Launchbury, 2017). However, developing symbol systems is knowledge-intensive and may lead to brittle systems, susceptible to the symbol grounding problem (Harnad, 1990).

We introduce a *Grounded Symbol System (GSS)* as the internal representation for systematic generalization, aiming to seamlessly integrate perception, syntax, and semantics, as depicted by Fig. 1. Formally, a GSS is a directed tree $T = < (x, s, v), e >$, with each node being a triplet consisting of grounded input $x$, an abstract symbol $s$, and its semantic meaning $v$. The edges $e$ represent semantic dependencies, with an edge $i \rightarrow j$ indicating node $i$'s meaning depends on node $j$.

Given the delicate nature and intensive effort required to create handcrafted symbol systems, the necessity of anchoring them in raw inputs and deriving their syntax and semantics directly from training data is paramount. This critical aspect is explored in depth in the subsequent sections.

### 3.2 MODEL: NEURAL-SYMBOLIC RECURSIVE MACHINE (NSR)

The NSR model, structured to induce a GSS from the training data, integrates three trainable modules (Fig. 2): neural perception for symbol grounding, dependency parsing to infer dependencies between symbols, and program synthesis to deduce semantic meanings. Given the absence of ground-truth GSS during training, these modules must be trained end-to-end without intermediate supervision. Below, we detail these modules and the end-to-end learning approach.

Figure 2: **Illustration of the inference and learning pipeline in NSR.**

**Neural Perception** This module converts raw input $x$ (*e.g.*, a handwritten expression) into a symbolic sequence $s$, represented by a list of indices. It addresses the perceptual variability of raw input signals, ensuring that each predicted token $w_i \in s$ accurately matches a specific segment of the input $x_i \in x$. Formally, this relationship is expressed as:

$$p(s|x; \theta_p) = \prod_i p(w_i|x_i; \theta_p) = \prod_i \text{softmax}(\phi(w_i, x_i; \theta_p)), \tag{1}$$

where $\phi(w_i, x_i; \theta_p)$ denotes a scoring function parameterized by a neural network with parameters $\theta_p$. The design of this neural network varies with the type of raw input and can be pre-trained, such as using a pre-trained convolutional neural network for processing image inputs.

**Dependency Parsing** To deduce the dependencies among symbols, we employ a transition-based neural dependency parser (Chen and Manning, 2014), a widely used method in natural language sentence parsing. Operating as a state machine, the parser identifies possible transitions to transform the input sequence into a dependency tree, iteratively applying predicted transitions until the parsing process is complete; see Fig. A1 for illustration. At each step, the parser predicts a transition based on the state representation, which includes the top elements of the stack and buffer, along with their direct descendants. A two-layer feed-forward neural network, given this state representation, determines the next transition. Formally, for an input sequence $s$, the parsing process is defined as:

$$p(e|s; \theta_s) = p(\mathcal{T}|s; \theta_s) = \prod_{t_i \in \mathcal{T}} p(t_i|c_i; \theta_s), \tag{2}$$

where $\theta_s$ are the parameters of the parser, $\mathcal{T} = \{t_1, t_2, ..., t_l\} \models e$ is the sequence of transitions that generates the dependency tree $e$, and $c_i$ is the state representation at step $i$. A greedy decoding strategy is employed to determine the most likely transition sequence for the given input sequence.

**Program Induction** Influenced by developments in program induction (Ellis et al., 2021; Balog et al., 2017; Devlin et al., 2017), we utilize *functional* programs to express the semantics of symbols, framing learning as a process of program induction. Symbolic programs offer enhanced generalizability, interpretability, and sample efficiency over purely statistical approaches. Formally, given input symbols $s$ and their dependencies $e$, the relationship is defined as:

$$p(v|e, s; \theta_l) = \prod_i p(v_i|s_i, \text{children}(s_i); \theta_l), \tag{3}$$

where $\theta_l$ denotes the set of induced programs for each symbol. Symbolic programs are utilized in practice, ensuring a deterministic reasoning process.

Deriving symbol semantics involves identifying a program that aligns with given examples, employing pre-defined primitives. Based on Peano axioms (Peano, 1889; Xie et al., 2021), we select a universal set of primitives, such as `0`, `inc`, `dec`, `==`, and `if`, proven to be sufficient for any symbolic function representation; see Sec. 3.4 for an in-depth discussion. To streamline the search and improve generalization, we include a minimal subset of Lisp primitives and the recursion primitive (Y-combinator (Peyton Jones, 1987)). DreamCoder (Ellis et al., 2021) is adapted for program induction; we modified it to accommodate noise in examples during the search.

**Model Inference** Fig. 2 depicts the model inference process, beginning with the neural perception module translating input $x$, such as a handwritten expression from Fig. 1 (3) HINT, into a sequence of symbols, $2 + 3 \times 9$. The dependency parsing module then structures this sequence into a dependency tree, for instance, $+ \rightarrow 2\times$ and $\times \rightarrow 3\,9$. Subsequently, the program induction module employs programs associated with each symbol to compute node values in a bottom-up fashion, yielding calculations like $3 \times 9 \Rightarrow 27$, followed by $2 + 27 \Rightarrow 29$.

## 3.3 LEARNING

The intermediate GSS in NSR is both latent and non-differentiable, making direct application of backpropagation unfeasible. Traditional approaches, such as policy gradient algorithms like REINFORCE (Williams, 1992), face difficulties with slow or inconsistent convergence (Liang et al., 2018; Li et al., 2020). Given the vast search space for GSS, a more efficient learning algorithm is imperative. Formally, for input $x$, intermediate GSS $T = <(x, s, v), e>$, and output $y$, the likelihood of observing $(x, y)$, marginalized over $T$, is expressed as:

$$p(y|x; \Theta) = \sum_T p(T, y|x; \Theta) = \sum_{s,e,v} p(s|x; \theta_p) p(e|s; \theta_s) p(v|s, e; \theta_l) p(y|v), \tag{4}$$

where maximizing the observed-data log-likelihood $L(x, y) = \log p(y|x)$ becomes the learning objective, from a maximum likelihood estimation viewpoint. The gradients of $L$ with respect to $\theta_p, \theta_s, \theta_l$ are as follows:

$$\begin{aligned}
\nabla_{\theta_p} L(x, y) &= \mathbb{E}_{T \sim p(T|x,y)}[\nabla_{\theta_p} \log p(s|x; \theta_p)], \\
\nabla_{\theta_s} L(x, y) &= \mathbb{E}_{T \sim p(T|x,y)}[\nabla_{\theta_s} \log p(e|s; \theta_s)], \\
\nabla_{\theta_l} L(x, y) &= \mathbb{E}_{T \sim p(T|x,y)}[\nabla_{\theta_l} \log p(v|s, e; \theta_l)],
\end{aligned} \tag{5}$$

where $p(T|x, y)$ denotes the posterior distribution of $T$ given $(x, y)$, which can be represented as:

$$p(T|x, y) = \frac{p(T, y|x; \Theta)}{\sum_{T'} p(T', y|x; \Theta)} = \begin{cases} 0, & \text{if } T \notin Q \\ \frac{p(T|x; \Theta)}{\sum_{T' \in Q} p(T'|x; \Theta)}, & \text{if } T \in Q \end{cases} \tag{6}$$

with $Q$ as the set of $T$ congruent with $y$.

Due to the computational challenge of taking expectations with respect to this posterior distribution, Monte Carlo sampling is utilized for approximation. This optimization strategy entails sampling a solution from the posterior distribution and iteratively updating each module through supervised training, circumventing the difficulty of sampling within a vast, sparsely populated space.

**Deduction-Abduction** The essence of our learning approach is an efficient sampling from the posterior distribution $p(T|x, y)$, achieved through the *deduction-abduction* algorithm (Alg. A1). For an instance $(x, y)$, we initially perform a greedy deduction from $x$ to obtain an initial GSS $T = <(x, \hat{s}, \hat{v}), \hat{e}>$. To align $T^*$ with the correct result $y$ during training, a top-down abduction search is employed, iterating over the neighbors of $T$ and adjusting across perception, syntax, and semantics; see Figs. 3 and A2 for more details. This search ceases upon finding a $T^*$ that yields $y$ or reaching a predetermined number of steps. Theoretically, this method acts as a Metropolis-Hastings sampler for $p(T|x, y)$, facilitating an efficient approach to model training (Li et al., 2020).

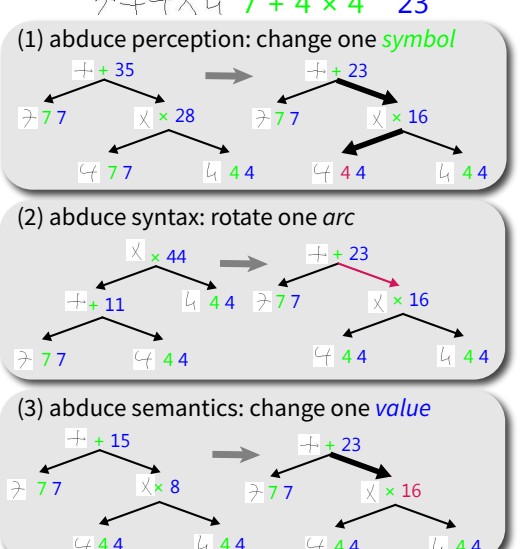

Figure 3: **Illustration of abduction in HINT over perception, syntax, and semantics.** Elements modified during abduction are emphasized in red.

## 3.4 EXPRESSIVENESS AND GENERALIZATION OF NSR

We now examine the characteristics of NSR, particularly its expressiveness and ability to systematically generalize, which are attributed to its inherent inductive biases.

**Expressiveness**  The capacity of NSR to represent a broad spectrum of seq2seq tasks is substantiated by theoretical analysis. The following theorem articulates the model's expressiveness.

**Theorem 3.1.** Given any finite dataset $D = \{(x^i, y^i)\}_{i=0}^N$, there exists an NSR that can express $D$ using only four primitives: $\{\texttt{0}, \texttt{inc}, \texttt{==}, \texttt{if}\}$.

The proof of this theorem involves constructing a specialized NSR that effectively "memorizes" all examples in $D$ through a comprehensive program:

$$\text{NSR}(x) = \texttt{if}(f_p(x)\texttt{==}0, y^0, \texttt{if}(f_p(x)\texttt{==}1, y^1, ...\texttt{if}(f_p(x)\texttt{==}N, y^N, \varnothing)...), \tag{7}$$

serving as a sophisticated lookup table constructed with $\{\texttt{if}, \texttt{==}\}$ and the indexing scheme provided by $\{\texttt{0}, \texttt{inc}\}$(proved by Lemmas C.1 and C.2). Given the universality of these four primitives across domains, NSR's ability to model a variety of seq2seq tasks is assured, offering improved transferability compared to preceding neural-symbolic models.

**Generalization**  The program outlined in Eq. (7), while guaranteeing perfect accuracy on the training dataset, lacks in generalization capacity. For effective generalization, it is essential to incorporate certain inductive biases that are minimal yet universally applicable across various domains. Inspired by seminal works in compositional generalization (Gordon et al., 2019; Chen et al., 2020; Xie et al., 2021; Zhang et al., 2022), we advocate for two critical inductive biases: *equivariance* and *compositionality*. Equivariance enhances the model's systematicity, enabling it to generalize concepts such as ``jump twice'' from examples like $\{$``run'', ``run twice'', ``jump''$\}$, whereas compositionality increases the model's ability to extend these concepts to longer sequences, *e.g.*, ``run and jump twice''.

The formal definitions of equivariance and compositionality are as follows:

**Definition 3.1** (*Equivariance*). For sets $\mathcal{X}$ and $\mathcal{Y}$, and a *permutation* group $\mathcal{P}$ with operations $T_p : \mathcal{X} \to \mathcal{X}$ and $T'_p : \mathcal{Y} \to \mathcal{Y}$, a mapping $\Phi : \mathcal{X} \to \mathcal{Y}$ is *equivariant* iff $\forall x \in \mathcal{X}, p \in \mathcal{P} : \Phi(T_p x) = T'_p \Phi(x)$.

**Definition 3.2** (*Compositionality*). For sets $\mathcal{X}$ and $\mathcal{Y}$, with composition operations $T_c : (\mathcal{X}, \mathcal{X}) \to \mathcal{X}$ and $T'_c : (\mathcal{Y}, \mathcal{Y}) \to \mathcal{Y}$, a mapping $\Phi : \mathcal{X} \to \mathcal{Y}$ is *compositional* iff $\exists T_c, T'_c, \forall x_1 \in \mathcal{X}, x_2 \in \mathcal{X} : \Phi(T_c(x_1, x_2)) = T'_c(\Phi(x_1), \Phi(x_2))$.

The three modules of NSR—neural perception (Eq. (1)), dependency parsing (Eq. (2)), and program induction (Eq. (3))—exhibit equivariance and compositionality, functioning as pointwise transformations based on their formulations. Models demonstrating exceptional compositional generalization, such as NeSS (Chen et al., 2020), LANE (Liu et al., 2020), and NSR, inherently possess these properties. This leads to our hypothesis regarding compositional generalization:

**Hypothesis 3.1.** A model achieving compositional generalization instantiates a mapping $\Phi : \mathcal{X} \to \mathcal{Y}$ that is inherently *equivariant* and *compositional*.

## 4 EXPERIMENTS

Our evaluation of the NSR's capabilities in systematic generalization extends across three distinct benchmarks: (i) SCAN (Lake and Baroni, 2018), focusing on the translation of natural language commands to action sequences; (ii) PCFG (Hupkes et al., 2020), aimed at predicting output sequences from string manipulation commands; and (iii) HINT (Li et al., 2023b), predicting outcomes of handwritten arithmetic expressions. Furthermore, we explore NSR's performance on a compositional machine translation task (Lake and Baroni, 2018) to validate its applicability to real-world scenarios.

### 4.1 SCAN

The SCAN dataset (Lake and Baroni, 2018) emerges as a critical benchmark for assessing machine learning models' systematic generalization capabilities. It challenges models to translate natural language directives into sequences of actions, simulating the movement of an agent within a grid-like environment.

**Evaluation Protocols**  Following established studies (Lake, 2019; Gordon et al., 2019; Chen et al., 2020), we assess NSR using four splits: (i) SIMPLE, where the dataset is randomly divided into

training and test sets; (ii) LENGTH, with training on output sequences up to 22 actions and testing on sequences from 24 to 48 actions; (iii) JUMP, training excluding the "jump" command mixed with other primitives, tested on combinations including "jump"; and (iv) AROUND RIGHT, excluding "around right" from training but testing on combinations derived from the separately trained "around" and "right."

**Baselines** NSR is compared against multiple models including (i) seq2seq (Lake and Baroni, 2018), (ii) CNN (Dessì and Baroni, 2019), (iii) Transformer (Csordás et al., 2021; Ontanón et al., 2022), (iv) equivariant seq2seq (Gordon et al., 2019)—a model that amalgamates convolutional operations with RNNs to attain local equivariance, and (v) NeSS (Chen et al., 2020)—a model that integrates a symbolic stack machine into a seq2seq framework.

**Results** The summarized results are presented in Tab. 1. Remarkably, both NSR and NeSS realize 100% accuracy on the splits necessitating systematic generalization. In contrast, the peak performance of other models on the LENGTH split barely reaches 20%. This stark discrepancy underscores the pivotal role and efficacy of symbolic components—specifically, the symbolic stack machine in NeSS and the GSS in NSR—in fostering systematic generalization.

Table 1: **Test accuracy across various splits of SCAN and PCFG.** The results of NeSS on PCFG are reported by modifying the source code from Chen et al. (2020) for PCFG.

| models | SCAN | | | | PCFG | | |
|---|---|---|---|---|---|---|---|
| | SIMPLE | JUMP | AROUND RIGHT | LENGTH | i.i.d. | systematicity | productivity |
| Seq2Seq (Lake and Baroni, 2018) | 99.7 | 1.2 | 2.5 | 13.8 | 79 | 53 | 30 |
| CNN (Dessì and Baroni, 2019) | 100.0 | 69.2 | 56.7 | 0.0 | 85 | 56 | 31 |
| Transformer (Csordás et al., 2021) | - | - | - | 20.0 | - | 96 | 85 |
| Transformer (Ontanón et al., 2022) | - | 0.0 | - | 19.6 | - | 83 | 63 |
| equivariant Seq2seq (Gordon et al., 2019) | 100.0 | 99.1 | 92.0 | 15.9 | - | - | - |
| NeSS (Chen et al., 2020) | 100.0 | 100.0 | 100.0 | 100.0 | $\approx 0$ | $\approx 0$ | $\approx 0$ |
| NSR (ours) | **100.0** | **100.0** | **100.0** | **100.0** | **100** | **100** | **100** |

While NeSS and NSR both manifest impeccable generalization on SCAN, their foundational principles are distinctly divergent.

1. NeSS necessitates an extensive reservoir of domain-specific knowledge to meticulously craft the components of the stack machine, encompassing stack operations and category predictors. Without the incorporation of category predictors, the efficacy of NeSS plummets to approximately 20% in 3 out of 5 runs. Contrarily, NSR adopts a modular architecture, minimizing reliance on domain-specific knowledge.

2. The training regimen for NeSS is contingent on a manually curated curriculum, coupled with specialized training protocols for latent category predictors. Conversely, NSR is devoid of any prerequisite for a specialized training paradigm.

Fig. 4 elucidates the syntax and semantics assimilated by NSR from the LENGTH split in SCAN. The dependency parser of NSR, exhibiting equivariance as elucidated in Sec. 3.4, delineates distinct permutation equivalent groups syntactically among the input words: {turn, walk, look, run, jump}, {left, right, opposite, around, twice, thrice}, and {and, after}. It is pivotal to note that no prior information regarding these groups is imparted—they are entirely a manifestation of the learning from the training data. This is in stark contrast to the provision of pre-defined equivariant groups (Gordon et al., 2019) or the explicit integration of a category induction procedure from execution traces (Chen et al., 2020). Within the realm of the learned programs, the program synthesizer of NSR formulates an index space for the target language and discerns the accurate programs to encapsulate the semantics of each source word.

## 4.2 PCFG

We further assess NSR on the PCFG dataset (Hupkes et al., 2020), a task where the model is trained to predict the output of a string manipulation command. The input sequences in PCFG are synthesized using a probabilistic context-free grammar, and the corresponding output sequences are formed by recursively executing the string edit operations delineated in the input sequences. The selection of input samples is designed to mirror the statistical attributes of a natural language corpus, including sentence lengths and parse tree depths.

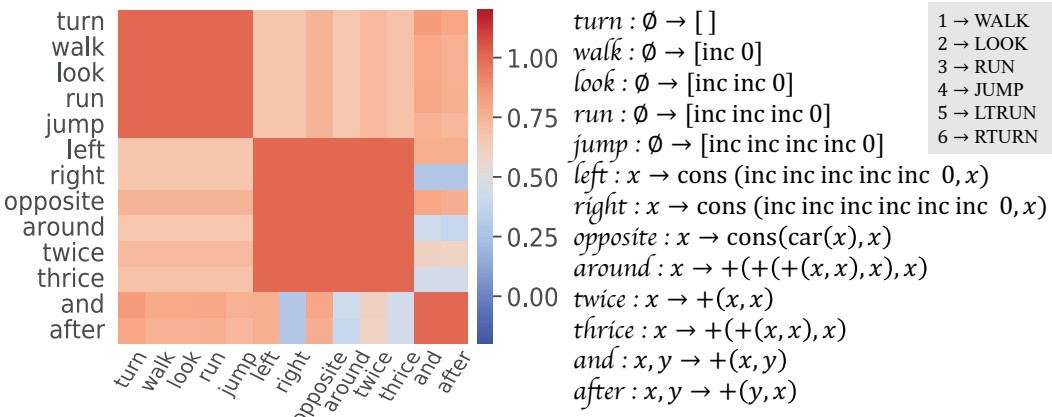

(a) Syntactic similarity amongst input words in NSR.     (b) Programs induced using NSR.

Figure 4: (a) **Syntactic similarity amongst input words in NSR trained on the LENGTH split in SCAN.** The similarity between word $i$ and word $j$ is quantified by the percentage of test samples where substituting $i$ with $j$, or vice versa, retains the dependencies as predicted by the dependency parser. (b) **Induced programs for input words using NSR.** Here, $x$ and $y$ represent the inputs, $\varnothing$ signifies empty inputs, `cons` appends an item to the beginning of a list, `car` retrieves the first item of a list, and + amalgamates two lists.

**Evaluation Protocols and Baselines**    The evaluation is conducted across the following splits: (i) i.i.d: where samples are randomly allocated for training and testing, (ii) systematicity: this split is designed to specifically assess the model's capability to interpret combinations of functions unseen during training, and (iii) productivity: this split tests the model's generalization to longer sequences, with training samples containing up to 8 functions and test samples having at least 9 functions. NSR is compared against (i) seq2seq (Lake and Baroni, 2018), (ii) CNN (Dessì and Baroni, 2019), (iii) Transformer (Csordás et al., 2021; Ontanón et al., 2022), and (iv) NeSS (Chen et al., 2020).

**Results**    The results are consolidated in Tab. 1. NSR demonstrates exemplary performance, achieving 100% accuracy across all PCFG splits and surpassing the prior best-performing model (Transformer) by 4% on the "systematicity" split and by 15% on the "productivity" split. Notably, while NeSS exhibits flawless accuracy on SCAN, it encounters total failure on PCFG. A closer examination of NeSS's training on PCFG reveals that its stack operations cannot represent PCFG's binary functions, and the trace search process is hindered by PCFG's extensive vocabulary and elongated sequences. Adapting NeSS to this context would necessitate substantial domain-specific modifications and extensive refinements to both the stack machine and the training methodology.

## 4.3   HINT

We also evaluate NSR on HINT (Li et al., 2023b), a task where the model predicts the integer result of a handwritten arithmetic expression, such as ( 3 + 2 ) ✕ 8 → 40, without any intermediate supervision. HINT is challenging due to the high variance and ambiguity in real handwritten images, the complexity of syntax due to parentheses, and the involvement of recursive functions in semantics. The dataset includes one training set and five test subsets, each assessing different aspects of generalization across perception, syntax, and semantics.

**Evaluation Protocols and Baselines**    Adhering to the protocols of Li et al. (2023b), we train models on a single training set and evaluate them on five test subsets: (i) "I": expressions seen in training but composed of unseen handwritten images. (ii) "SS": unseen expressions, but their lengths and values are within the training range. (iii) "LS": expressions are longer than those in training, but their values are within the same range. (iv) "SL": expressions have larger values, and their lengths are the same as training. (v) "LL": expressions are longer, and their values are bigger than those in training. A prediction is deemed correct only if it exactly matches the ground truth. We compare NSR against several baselines including seq2seq models like GRU, LSTM, and Transformer as reported by Li et al. (2023b), and NeSS (Chen et al., 2020), with each model utilizing a ResNet-18 as the image encoder.

**Results**    The results are summarized in Tab. 2. NSR surpasses the state-of-the-art model, Transformer, by approximately 23%. The detailed results reveal that this improvement is primarily due to better extrapolation in syntax and semantics, with NSR elevating the accuracy on the "LL" subset

Table 2: **Test accuracy on HINT.** Results for GRU, LSTM, and Transformer are directly cited from Li et al. (2023b). NeSS results are obtained by adapting its source code to HINT.

| Model | Symbol Input | | | | | | Image Input | | | | | |
|---|---|---|---|---|---|---|---|---|---|---|---|---|
| | I | SS | LS | SL | LL | Avg. | I | SS | LS | SL | LL | Avg. |
| GRU | 76.2 | 69.5 | 42.8 | 10.5 | 15.1 | 42.5 | 66.7 | 58.7 | 33.1 | 9.4 | 12.8 | 35.9 |
| LSTM | 92.9 | 90.9 | 74.9 | 12.1 | 24.3 | 58.9 | 83.9 | 79.7 | 62.0 | 11.2 | 21.0 | 51.5 |
| Transformer | 98.0 | 96.8 | 78.2 | 11.7 | 22.4 | 61.5 | 88.4 | 86.0 | 62.5 | 10.9 | 19.0 | 53.1 |
| NeSS | $\approx 0$ | $\approx 0$ | $\approx 0$ | $\approx 0$ | $\approx 0$ | $\approx 0$ | - | - | - | - | - | - |
| NSR (ours) | **98.0** | **97.3** | **83.7** | **95.9** | **77.6** | **90.1** | **88.5** | **86.2** | **67.1** | **83.2** | **58.2** | **76.0** |

from 19.0% to 58.2%. The gains on the "I" and "SS" subsets are more modest, around 2%. For a more detailed insight into NSR's predictions on HINT, refer to Fig. A3. Similar to its performance on PCFG, NeSS fails on HINT, underscoring the challenges discussed in Sec. 4.2.

## 4.4 COMPOSITIONAL MACHINE TRANSLATION

In order to assess the applicability of NSR to real-world scenarios, we conduct a proof-of-concept experiment on a compositional machine translation task, specifically the English-French translation task, as described by Lake and Baroni (2018). This task has been a benchmark for several studies (Li et al., 2019; Chen et al., 2020; Kim, 2021) to validate the efficacy of their proposed methods in more realistic and complex domains compared to synthetic tasks like SCAN and PCFG. The complexity and ambiguity of the rules in this translation task are notably higher.

We utilize the publicly available data splits provided by Li et al. (2019). The training set comprises 10,000 English-French sentence pairs, with English sentences primarily initiating with phrases like "I am," "you are," and their respective contractions. Uniquely, the training set includes 1,000 repetitions of the sentence pair ("I am daxy," "je suis daxiste"), introducing the pseudoword "daxy." The test set, however, explores 8 different combinations of "daxy" with other phrases, such as "you are not daxy," which are absent from the training set.

**Results** Tab. 3 presents the results of the compositional machine translation task. We compare NSR with Seq2Seq (Lake and Baroni, 2018) and NeSS (Chen et al., 2020). It is noteworthy that two distinct French translations for "you are" are prevalent in the training set; hence, both are deemed correct in the test set. NSR, akin to NeSS, attains 100% generalization accuracy on this task, demonstrating its potential applicability to real-world tasks characterized by diverse and ambiguous rules.

Table 3: **Accuracy on compositional machine translation.**

| Model | Accuracy |
|---|---|
| Seq2Seq | 12.5 |
| Transformer | 14.4 |
| NeSS | 100.0 |
| NSR (ours) | 100.0 |

## 5 CONCLUSION AND DISCUSSION

We introduced NSR, a model capable of learning Grounded Symbol System from data to facilitate systematic generalization. The Grounded Symbol System offers a generalizable and interpretable representation, allowing a principled amalgamation of perception, syntax, and semantics. NSR employs a modular design, incorporating the essential inductive biases of equivariance and compositionality in each module to realize compositional generalization. We developed a probabilistic learning framework and introduced a novel deduction-abduction algorithm to enable the efficient training of NSR without GSS supervision. NSR has demonstrated superior performance across diverse domains, including semantic parsing, string manipulation, arithmetic reasoning, and compositional machine translation.

**Limitations** While NSR has shown impeccable accuracy on a conceptual machine translation task, we foresee challenges in its deployment for real-world tasks due to (i) the presence of noisy and abundant concepts, which may enlarge the space of the grounded symbol system and potentially decelerate the training of NSR, and (ii) the deterministic nature of the functional programs in NSR, limiting its ability to represent probabilistic semantics inherent in real-world tasks, such as the existence of multiple translations for a single sentence. Addressing these challenges remains a subject for future research.

**Acknowledgment** We thank the anonymous reviewers for their valuable feedback and constructive suggestions. Their insights have contributed significantly to improving the quality and clarity of our work. Additionally, we sincerely appreciate the area chairs and program chairs for their efforts in organizing the review process and providing valuable guidance throughout the submission. We would like to thank NVIDIA for their generous support of GPUs and hardware. This work is supported in part by the National Science and Technology Major Project (2022ZD0114900) and the Beijing Nova Program.

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

# A    Model Details

**Dependency Parsing**    Fig. A1 illustrate the process of parsing an arithmetic expression via the dependency parser. Formally, a *state* $c = (\alpha, \beta, A)$ in the dependency parser consists of a *stack* $\alpha$, a *buffer* $\beta$, and a set of *dependency arcs* $A$. The initial state for a sequence $s = w_0 w_1 ... w_n$ is $\alpha = [\texttt{Root}], \beta = [w_0 w_1 ... w_n], A = \varnothing$. A state is regarded as terminal if the buffer is empty and the stack only contains the node $\texttt{Root}$. The parse tree can be derived from the dependency arcs $A$. Let $\alpha_i$ denote the $i$-th top element on the stack, and $\beta_i$ the $i$-th element on the buffer. The parser defines three types of transitions between states:

- LEFT-ARC: add an arc $\alpha_1 \rightarrow \alpha_2$ to $A$ and remove $\alpha_2$ from the stack $\alpha$. Precondition: $|\alpha| \geqslant 2$.
- RIGHT-ARC: add an arc $\alpha_2 \rightarrow \alpha_1$ to $A$ and remove $\alpha_1$ from the stack $\alpha$. Precondition: $|\alpha| \geqslant 2$.
- SHIFT: move $\beta_1$ from the buffer $\beta$ to the stack $\alpha$. Precondition: $|\beta| \geqslant 1$.

The goal of the parser is to predict a transition sequence from an initial state to a terminal state. The parser predicts one transition from $\mathcal{T} = \{\text{LEFT-ARC}, \text{RIGHT-ARC}, \text{SHIFT}\}$ at a time, based on the current state $c = (\alpha, \beta, A)$. The state representation is constructed from a local window and contains following three elements: (i) The top three words on the stack and buffer: $\alpha_i, \beta_i, i = 1, 2, 3$; (ii) The first and second leftmost/rightmost children of the top two words on the stack: $lc_1(\alpha_i), rc_1(\alpha_i), lc_2(\alpha_i), rc_2(\alpha_i), i = 1, 2$; (iii) The leftmost of leftmost/rightmost of rightmost children of the top two words on the stack: $lc_1(lc_1(\alpha_i)), rc_1(rc_1(\alpha_i)), i = 1, 2$. We use a special $\texttt{Null}$ token for non-existent elements. Each element in the state representation is embedded to a $d$-dimensional vector $e \in R^d$, and the full embedding matrix is denoted as $E \in R^{|\Sigma| \times d}$, where $\Sigma$ is the concept space. The embedding vectors for all elements in the state are concatenated as its representation: $c = [e_1 \; e_2 ... e_n] \in R^{nd}$. Given the state representation, we adopt a two-layer feed-forward neural network to predict the transition.

**Program Induction**    Program induction, *i.e.*, synthesizing programs from input-output examples, was one of the oldest theoretical frameworks for concept learning within artificial intelligence (Solomonoff, 1964). Recent advances in program induction focus on training neural networks to guide the program search (Kulkarni et al., 2015; Lake et al., 2015; Balog et al., 2017; Devlin et al., 2017; Ellis et al., 2018a;b). For example, Balog et al. (2017) train a neural network to predict properties of the program that generated the outputs from the given inputs and then use the neural network's predictions to augment search techniques from the programming languages community. Ellis et al. (2021) released a neural-guided program induction system, *DreamCoder*, which can efficiently discover interpretable, reusable, and generalizable programs across a wide range of domains, including both classic inductive programming tasks and creative tasks such as drawing pictures and building scenes. DreamCoder adopts a "wake-sleep" Bayesian learning algorithm to extend program space with new symbolic abstractions and train the neural network on imagined and replayed problems.

To learn the semantics of a symbol $c$ from a set of examples $D_c$ is to find a program $\rho_c$ composed from a set of primitives $L$, which maximizes the following objective:

$$\max_{\rho} p(\rho | D_c, L) \propto p(D_c | \rho) \, p(\rho | L), \tag{A1}$$

where $p(D_c | \rho)$ is the likelihood of the program $\rho$ matching $D_c$, and $p(\rho | L)$ is the prior of $\rho$ under the program space defined by the primitives $L$. Since finding a globally optimal program is usually intractable, the maximization in Eq. (A1) is approximated by a stochastic search process guided by a neural network, which is trained to approximate the posterior distribution $p(\rho | D_c, L)$. We refer the readers to DreamCoder (Ellis et al., 2021)[1] for more technical details.

---

[1] https://github.com/ellisk42/ec

# B    LEARNING

**Derivation of Eq. (5)**    Take the derivative of $L$ w.r.t. $\theta_p$,

$$\nabla_{\theta_p} L(x, y) = \nabla_{\theta_p} \log p(y|x) = \frac{1}{p(y|x)} \nabla_{\theta_p} p(y|x)$$

$$= \sum_T \frac{p(T, y|x; \Theta)}{\sum_{T'} p(T', y|x; \Theta)} \nabla_{\theta_p} \log p(s|x; \theta_p) \qquad (A2)$$

$$= \mathbb{E}_{T \sim p(T|x, y)} [\nabla_{\theta_p} \log p(s|x; \theta_p)].$$

Similarly, for $\theta_s, \theta_l$, we have

$$\nabla_{\theta_s} L(x, y) = \mathbb{E}_{T \sim p(T|x, y)} [\nabla_{\theta_s} \log p(e|s; \theta_s)],$$
$$\nabla_{\theta_l} L(x, y) = \mathbb{E}_{T \sim p(T|x, y)} [\nabla_{\theta_l} \log p(v|s, e; \theta_l)], \qquad (A3)$$

**Deduction-Abduction**    Alg. A1 describes the procedure for learning NSR by the proposed deduction-abduction algorithm. Fig. 3 illustrates the one-step abduction over perception, syntax, and semantics in HINT and Fig. A2 visualizes a concrete example to illustrate the deduction-abduction process. It is similar for SCAN and PCFG.

# C    EXPRESSIVENESS AND GENERALIZATION OF NSR

**Expressiveness**

**Lemma C.1.**  Given a finite unique set of $\{x^i : i = 0, ..., N\}$, there exists a sufficiently capable neural network $f_p$ such that: $\forall x^i, f_p(x^i) = i$.

This lemma asserts the existence of a neural network capable of mapping every element in a finite set to a unique index, *i.e.*, $x^i \rightarrow i$, as supported by (Hornik et al., 1989; Lu et al., 2017). The parsing process in this scenario is straightforward, given that every input is mapped to a singular token.

**Lemma C.2.**  Any index space can be constructed from the primitives $\{0, \text{inc}\}$.

This lemma is grounded in the fact that all indices are natural numbers, which can be recursively defined by $\{0, \text{inc}\}$, allowing the creation of indices for both inputs and outputs.

**Generalization**    Equivariance and compositionality are formalized utilizing group theory, following the approaches of Gordon et al. (2019) and Zhang et al. (2022). A discrete group $\mathcal{G}$ comprises elements $\{g_1, ..., g_{|\mathcal{G}|}\}$ and a binary group operation "·", adhering to group axioms (closure, associativity, identity, and invertibility). Equivariance is associated with a *permutation* group $\mathcal{P}$, representing permutations of a set $\mathcal{X}$. For compositionality, a *composition* operation $\mathcal{C}$ is considered, defining $T_c : (\mathcal{X}, \mathcal{X}) \rightarrow \mathcal{X}$.

The three modules of NSR—neural perception (Eq. (1)), dependency parsing (Eq. (2)), and program induction (Eq. (3))—exhibit equivariance and compositionality, functioning as pointwise transformations based on their formulations. Eqs. (1) to (3) demonstrate that in all three modules of the NSR system, the joint distribution is factorized into a product of several independent terms. This factorization process makes the modules naturally adhere to the principles of equivariance and recursiveness, as outlined in Definitions 3.1 and 3.2.

# D    EXPERIMENTS

## D.1    EXPERIMENTAL SETUP

For tasks taking symbols as input (*i.e.*, SCAN and PCFG), the perception module is not required in NSR; For the task taking images as input, we adopt ResNet-18 as the perception module, which is pre-trained unsupervisedly (Van Gansbeke et al., 2020) on handwritten images from the training set. In the dependency parser, the token embeddings have a dimension of 50, the hidden dimension of the transition classifier is 200, and we use a dropout of 0.5. For the program induction, we adopt the

default setting in DreamCoder (Ellis et al., 2021). For learning NSR, both the ResNet-18 and the dependency parser are trained by the Adam optimizer (Kingma and Ba, 2015) with a learning rate of $10^{-4}$. NSR are trained for 100 epochs for all datasets.

**Compute** All training can be done using a single NVIDIA GeForce RTX 3090Ti under 24 hours.

### D.2 ABLATION STUDY

To explore how well the individual modules of NSR are learned, we perform an ablation study on HINT to analyze the performance of each module of NSR. Specifically, along with the final results, the HINT dataset also provides the symbolic sequences and parse trees for evaluation. For Neural Perception, we report the accuracy of classifying each symbol. For Dependency parsing, we report the accuracy of attaching each symbol to its correct parent, given the ground-truth symbol sequence as the input. For Program Induction, we report the accuracy of the final results, given the ground-truth symbol sequence and parse tree.

Overall, each module achieves high accuracy, as shown in Tab. A1. For Neural Perception, most errors come from the two parentheses, ”(” and ”)”, because they are visually similar. For Dependency Parsing, we analyze the parsing accuracies for different concept groups: digits (100%), operators (95.85%), and parentheses (64.28%). The parsing accuracy of parentheses is much lower than those of digits and operators. We think this is because, as long as digits and operators are correctly parsed in the parsing tree, where to attach the parentheses does not influence the final results because parentheses have no semantic meaning. For Program Induction, we can manually verify that the induced programs (Fig. 4) have correct semantics. The errors are caused by exceeding the recursion limit when calling the program for multiplication. The above analysis is also verified by the qualitative examples in Fig. A3.

### D.3 QUALITATIVE EXAMPLES

Figs. A3 and A4 show several examples of the NSR predictions on SCAN and HINT.

Fig. A5 illustrates the evolution of semantics along the training of NSR in HINT. This pattern is highly in accordance with how children learn arithmetic in developmental psychology (Carpenter et al., 1999): The model first masters the semantics of digits as *counting*, then learns $+$ and $-$ as recursive counting, and finally figures out how to define $\times$ and $\div$ based on $+$ and $-$. Crucially, $\times$ and $\div$ are impossible to be correctly learned before mastering $+$ and $-$. The model is endowed with such an incremental learning capability since the program induction module allows the semantics of concepts to be built compositionally from those learned earlier (Ellis et al., 2021).

| ID | Stack Buffer | Transition | Dependency |
|----|--------------|------------|------------|
| 0 | $3 + 4 \times 2$ | Shift | |
| 1 | $3 + 4 \times 2$ | Shift | |
| 2 | $3 + 4 \times 2$ | Left-Arc | $3 \leftarrow +$ |
| 3 | $+ 4 \times 2$ | Shift | |
| 4 | $+ 4 \times 2$ | Shift | |
| 5 | $+ 4 \times 2$ | Left-Arc | $4 \leftarrow \times$ |
| 6 | $+ \times 2$ | Shift | |
| 7 | $+ \times 2$ | Right-Arc | $\times \rightarrow 2$ |
| 8 | $+ \times$ | Right-Arc | $+ \rightarrow \times$ |

Start: $\sigma = [\text{ROOT}]$, $\beta = w_1, ..., w_n$, $A = \emptyset$

1. Shift $\quad\quad \sigma, w_i|\beta, A \Rightarrow \sigma|w_i, \beta, A$
2. Left-Arc$_r$ $\quad \sigma|w_i|w_j, \beta, A \Rightarrow \sigma|w_j, \beta, A \cup \{r(w_j,w_i)\}$
3. Right-Arc$_r$ $\quad \sigma|w_i|w_j, \beta, A \Rightarrow \sigma|w_i, \beta, A \cup \{r(w_i,w_j)\}$

Finish: $\sigma = [w]$, $\beta = \emptyset$

$$3 + 4 \times 2$$

Figure A1: **Applying the transition-based dependency parser to an example of HINT.** It is similar for SCAN and PCFG.

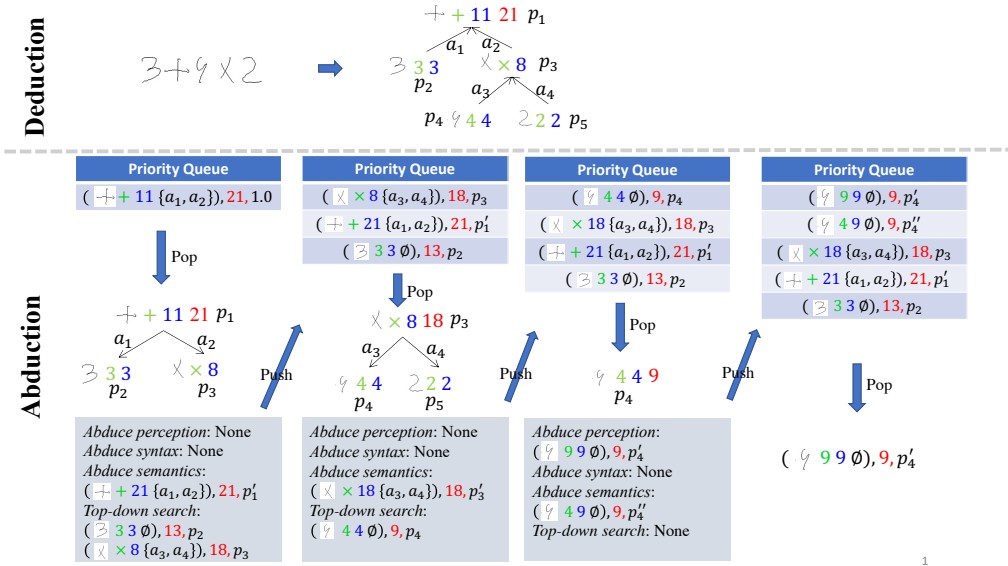

Figure A2: **An illustration of the deduction-abduction process for an example of HINT.** Given a handwritten expression, the system performs a greedy deduction to propose an initial solution, generating a wrong result. In abduction, the root node, paired with the ground-truth result, is first pushed to the priority queue. The abduction over perception, syntax, and semantics is performed on the popped node to generate possible revisions. A top-down search is also applied to propagate the expected value to its children. All possible revisions are then pushed into the priority queue. This process is repeated until we find the most likely revision for the initial solution.

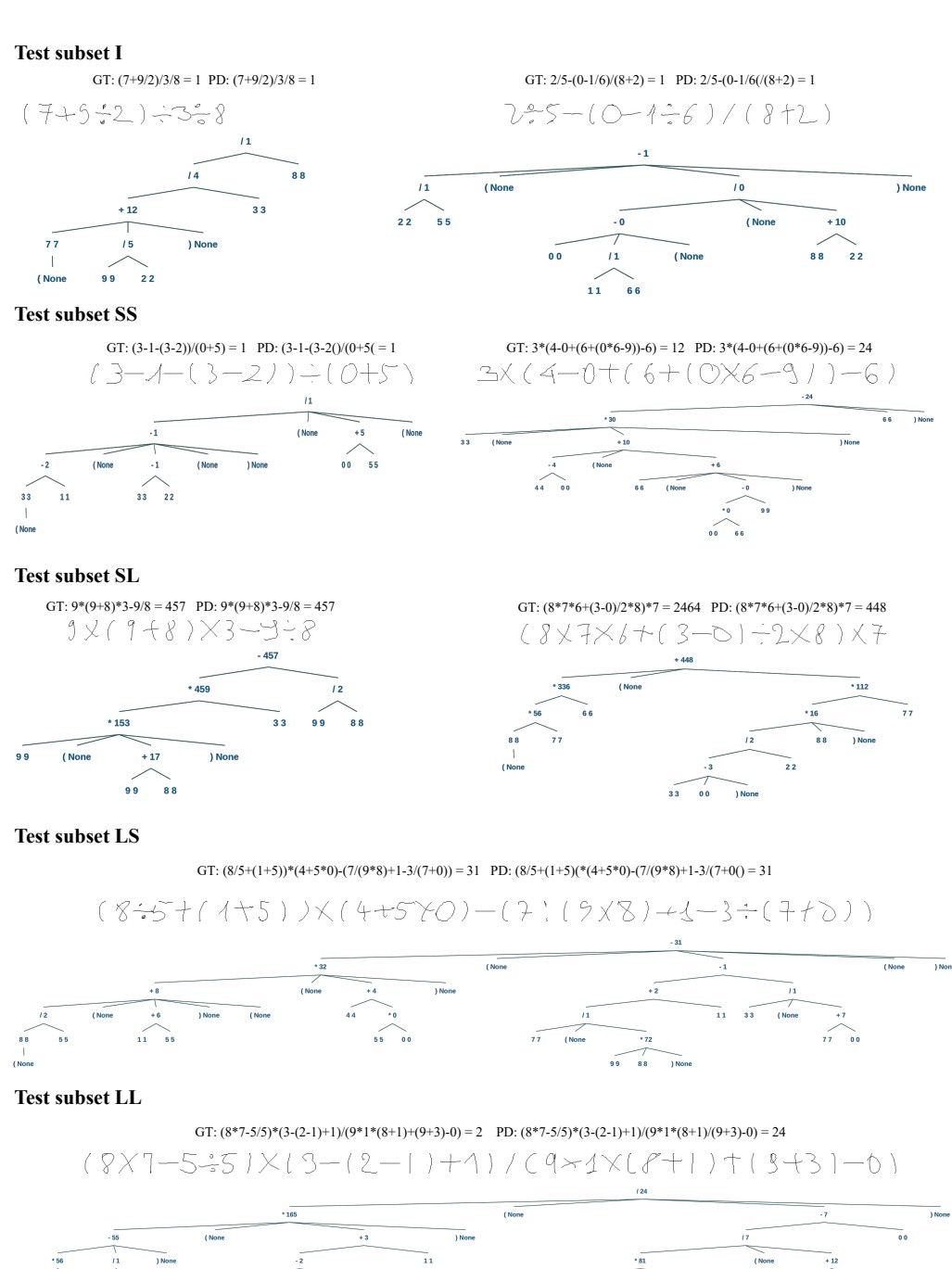

Figure A3: **Examples of NSR predictions on the test set of HINT.** "GT" and "PD" denote "ground-truth" and "prediction," respectively. Each node in the tree is a tuple of (symbol, value).

run around left twice and run around right

and 11 [LTURN, RUN] * 8 + [RTURN, RUN] * 4

twice 9 [LTURN, RUN] * 8          around 8 [RTURN, RUN] * 4

around 8 [LTURN,RUN] * 4          right 6 [RTURN, RUN]

left 5 [LTURN, RUN]          run 3 [RUN]

run 3 [RUN]

walk opposite right thrice after look around left twice

after 12 [LTURN, LOOK] * 8 + [RTURN, RTURN, WALK] * 3

thrice 10 [RTURN, RTURN, WALK] * 3          twice 9 [LTURN, LOOK] * 8

opposite 7 [RTURN, RTURN, WALK]          around 8 [LTURN, LOOK] * 4

right 6 [RTURN, WALK]          left 5 [LTURN, LOOK]

walk 1 [WALK]          look 2 [LOOK]

Figure A4: **Examples of NSR predictions on the test set of the SCAN LENGTH split.** We use * (repeating the list) and + (concatenating two lists) to shorten the outputs for easier interpretation.

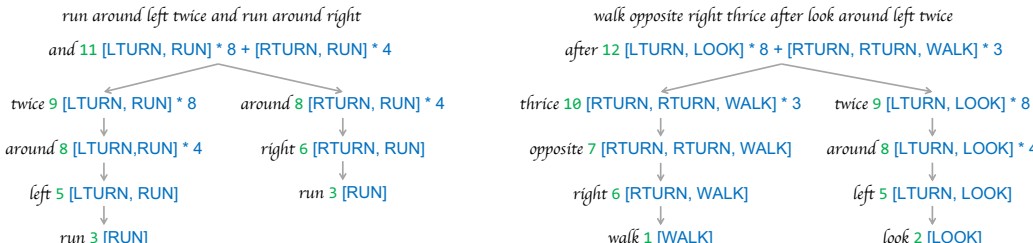

| | master counting | master + and − | | master × and ÷ | # Training epochs |
|---|---|---|---|---|---|
| 0: Null | 0: 0 | 0: 0 | | 0: 0 | |
| 1: Null | 1: inc 0 | 1: inc 0 | | 1: inc 0 | |
| 2: Null | 2: inc inc 0 | 2: inc inc 0 | | 2: inc inc 0 | |
| ... | ... | ... | | ... | |
| 9: Null | 9: inc inc ... inc 0 | 9: inc inc ... inc 0 | | 9: inc inc ... inc 0 | |
| +: Null | +: Null | +: if $(y == 0, x, +(\text{inc } x, \text{dec } y))$ | | +: if $(y == 0, x, (\text{inc } x) + (\text{dec } y))$ | |
| −: Null | −: Null | −: if $(y == 0, x, +(\text{dec } x, \text{dec } y))$ | | −: if $(y == 0, x, (\text{dec } x) + (\text{dec } y))$ | |
| ×: Null | ×: Null | ×: if $(y == 0, y, x)$ | | ×: if $(x == 0, 0, y \times (\text{dec } x) + y)$ | |
| ÷: Null | ÷: Null | ÷: if $(y == \text{inc } 0, x, \text{if } (x == 0, x, \text{inc inc } 0))$ | | ÷: if $(x == 0, 0, \text{inc } ((x - y) \div y))$ | |

Figure A5: **The evolution of learned programs in NSR for HINT.** The recursive programs in DreamCoder are represented by lambda calculus (a.k.a. $\lambda$-calculus) with Υ-combinator. Here, we translate the induced programs into pseudo code for easier interpretation. Note that there might be different yet functionally-equivalent programs to represent the semantics of a symbol; we only visualize a plausible one here.

Table A1: **Accuracy of the individual modules of NSR on the HINT dataset.**

| Module | Neural Perception | Dependency Parsing | Program Induction |
|---|---|---|---|
| **Accuracy** | 93.51 | 88.10 | 98.47 |

Table A2: **The test accuracy on different splits of SCAN and PCFG.** The results of NeSS on PCFG are reported by adapting the source code from Chen et al. (2020) on PCFG. Reported accuracy (%) is the average of 5 runs with standard deviation if available.

| models | SCAN | | | | PCFG | | |
|---|---|---|---|---|---|---|---|
| | SIMPLE | JUMP | AROUND RIGHT | LENGTH | i.i.d. | systematicity | productivity |
| Seq2Seq (Lake and Baroni, 2018) | 99.7 | 1.2 | 2.5 | 13.8 | 79 | 53 | 30 |
| CNN (Dessì and Baroni, 2019) | 100.0±0.0 | 69.2±8.2 | 56.7±10.2 | 0.0±0.0 | 85 | 56 | 31 |
| Transformer (Csordás et al., 2021) | - | - | - | 20.0 | - | 96±1 | 85±1 |
| Transformer (Ontanón et al., 2022) | - | 0.0 | - | 19.6 | - | 83 | 63 |
| equivariant Seq2seq (Gordon et al., 2019) | 100.0 | 99.1±0.04 | 92.0±0.24 | 15.9±3.2 | - | - | - |
| NeSS (Chen et al., 2020) | 100.0 | 100.0 | 100.0 | 100.0 | ≈0 | ≈0 | ≈0 |
| NSR (ours) | **100.0±0.0** | **100.0±0.0** | **100.0±0.0** | **100.0±0.0** | **100±0** | **100±0** | **100±0** |

Table A3: **The test accuracy on HINT.** We directly cite the results of GRU, LSTM, and Transformer from Li et al. (2023b). The results of NeSS are reported by adapting its source code on HINT. Reported accuracy (%) is the median and standard deviation of 5 runs.

| Model | Symbol Input | | | | | | Image Input | | | | | |
|---|---|---|---|---|---|---|---|---|---|---|---|---|
| | I | SS | LS | SL | LL | Avg. | I | SS | LS | SL | LL | Avg. |
| GRU | 76.2±0.6 | 69.5±0.6 | 42.8±1.5 | 10.5±0.2 | 15.1±1.2 | 42.5±0.7 | 66.7±2.0 | 58.7±2.2 | 33.1±2.7 | 9.4±0.3 | 12.8±1.0 | 35.9±1.6 |
| LSTM | 92.9±1.4 | 90.9±1.1 | 74.9±1.5 | 12.1±0.2 | 24.3±0.3 | 58.9±0.7 | 83.9±0.9 | 79.7±0.8 | 62.0±2.5 | 11.2±0.1 | 21.0±0.8 | 51.5±1.0 |
| Transformer | 98.0±0.3 | 96.8±0.6 | 78.2±2.9 | 11.7±0.3 | 22.4±1.1 | 61.5±0.9 | 88.4±1.3 | 86.0±1.3 | 62.5±4.1 | 10.9±0.2 | 19.0±1.0 | 53.1±1.6 |
| NeSS | ≈0 | ≈0 | ≈0 | ≈0 | ≈0 | ≈0 | - | - | - | - | - | - |
| NSR (ours) | **98.0±0.2** | **97.3±0.5** | **83.7±1.2** | **95.9±4.6** | **77.6±3.1** | **90.1±2.7** | **88.5±1.0** | **86.2±0.9** | **67.1±2.4** | **83.2±3.9** | **58.2±3.3** | **76.0±2.6** |

---

**Algorithm A1:** Learning by Deduction-Abduction

---

**Input** : Training set $D = (x_i, y_i) : i = 1, 2, ..., N$

**Output** : $\theta_p^{(T)}, \theta_s^{(T)}, \theta_l^{(T)}$

1   **Initial Module**: perception $\theta_p^{(0)}$, syntax $\theta_s^{(0)}$, semantics $\theta_l^{(0)}$

2   **for** $t \leftarrow 0$ *to* $T$ **do**

3      Buffer $\mathcal{B} \leftarrow \varnothing$

4      **foreach** $(x, y) \in D$ **do**

5         $T \leftarrow \text{DEDUCE}(x, \theta_p^{(t)}, \theta_s^{(t)}, \theta_l^{(t)})$

6         $T^* \leftarrow \text{ABDUCE}(T, y)$

7         $\mathcal{B} \leftarrow \mathcal{B} \cup T^*$

8      $\theta_p^{(t+1)}, \theta_s^{(t+1)}, \theta_l^{(t+1)} \leftarrow \text{learn}(\mathcal{B}, \theta_p^{(t)}, \theta_s^{(t)}, \theta_l^{(t)})$

9   **return** $\theta_p^{(T)}, \theta_s^{(T)}, \theta_l^{(T)}$

10

11   **Function** DEDUCE $(x, \theta_p, \theta_s, \theta_l)$:

12      Sample $\hat{s} \sim p(s|x; \theta_p), \hat{e} \sim p(e|\hat{s}; \theta_s), \hat{v} = f(\hat{s}, \hat{e}; \theta_l)$

13      **return** $T = \langle (x, \hat{s}, \hat{v}), \hat{e} \rangle$

14

15   **Function** ABDUCE $(T, y)$:

16      $Q \leftarrow \text{PriorityQueue}()$

17      $Q.push(\text{root}(T), y, 1.0)$

18      **while** $Q$ *is not empty* **do**

19         $A, y_A, p \leftarrow Q.pop()$

20         $A \leftarrow (x, w, v, arcs)$

21         **if** $A.v == y_A$ **then**

22            **return** $T(A)$

          // Abduce perception

23         **foreach** $w' \in \Sigma$ **do**

24            $A' \leftarrow A(w \rightarrow w')$

25            **if** $A'.v == y_A$ **then**

26               $Q.push(A', y_A, p(A'))$

          // Abduce syntax

27         **foreach** $arc \in arcs$ **do**

28            $A' \leftarrow \text{rotate}(A, arc)$

29            **if** $A'.v == y_A$ **then**

30               $Q.push(A', y_A, p(A'))$

          // Abduce semantics

31         $A' \leftarrow A(v \rightarrow y_A)$

32         $Q.push(A', y_A, p(A'))$

          // Top-down search

33         **foreach** $B \in children(A)$ **do**

34            $y_B \leftarrow \text{Solve}(B, A, y_A | \theta_l(A.w))$

35            $Q.push(B, y_B, p(B))$

36

---

