# OpenReview forum: "Neural-Symbolic Recursive Machine for Systematic Generalization"
_ICLR.cc/2024/Conference — ICLR 2024 poster_

### Official Review · Reviewer_WJqb · 2023-10-18

**Soundness:** 3 good
**Presentation:** 3 good
**Contribution:** 3 good
**Rating:** 8
**Confidence:** 4

**Summary:**

This work introduces a Neuro-symbolic approach capable of strong systematic generalization: NeuralSymbolic Recursive Machines (NSR).
The method is made of 3 modules:
1. a neural net perception module mapping raw input to grounded symbols. This can be a pre-trained CNN or transformer for Images and Text.
2. a dependency parser to infer dependencies between grounded symbols in a structured syntax tree, termed Grounded Symbol System (GSS)
3. a program synthesizer that deduces semantic meanings to a given symbol based on its neighborhood in the GSS tree.

To train this system with simple input-output (x, y) pairs and without any external expert knowledge or supervision for the GSS, the authors introduced a probabilistic learning method based on deduction-abduction: start with a greedy decoded and incorrect GSS tree, then refine step by step by looking at the potential neighbouring trees, until accurate results are obtained.
Monte-Carlo sampling is done to sample potential trees.

The method is tested on three synthetic tasks (SCAN, PCFG, Hint) and a compositional machine translation task. In SCAN, PCFG, and compositional machine translation, NSR obtains 100% accuracy. On Hint, NSR beats all baselines including vanilla Transformers.

**Strengths:**

This paper proposes a method that unifies the connectionism and the symbolism views of AI. While the attempt has been made multiple times in the past, the proposed method seems original and novel, although additional references could be cited (see the minor suggestions in the Weaknesses section)

The proposed method is well presented and the paper is easy to read. Figures 3 and 4 in particular made the content of the paper easier to understand and helped gain an intuition on what the method learns.
Experiments clearly present the strength of the proposed approach over previous baselines.

Eventually, this paper addresses an important challenge of current neural architectures: systematic generalization, making this work significant.

**Weaknesses:**

1. All results are comparing the proposed method NSR with various neural architectures and only one neuro-symbolic method: NeSS. The fact that NeSS performs 0% in 2 tasks and 100% in the other two makes it a weak comparing point (and also suggests that NeSS behaves more like a symbolic model than a neuro-symbolic one: it's all or nothing in terms of performance). I would suggest the authors provide at least 1 other neuro-symbolic method to compare against to make the results more significant. It is very clear that the proposed method outperforms vanilla neural methods such as Transformers, which is not surprising given the nature of the tasks being used, but it is less clear if the proposed method is significantly better than previous neuro-symbolic methods that also do not require additional training signal. The work from Minervini et. al. on Greedy Theorem Provers, or other variants could potentially be used as a baseline for some of these tasks.

2. Another weakness of this paper is the ambiguous explanation of how the search for a GSS tree is terminated. Section 1 states that the search for a tree runs "_until the accurate result is obtained_", and Section 3.2 doesn't detail this point (or at least not very well). The authors should better define this stop criterion in order to better understand its limitation: what does it mean for the resulting tree to be "accurate"? Could the method settle on an "almost correct" syntactic tree to save time? and what would the effect of that be on performance?

3. Eventually, at the end of Section 3, the authors state that the three modules of NSR exhibit equivariance and recursiveness. It would be beneficial to explain why this claim is true and provide additional evidence about it.

The following are minor suggestions:

4.  In Table 3, for the task of compositional translation, it would be interesting to also evaluate the performance of a vanilla transformer like in the previous tables.

5. the work could benefit from a discussion about previous neuro-symbolic works such as Neural Theorem Provers (NTPs) and Greedy NTPs: "_Differentiable Reasoning on Large Knowledge Bases and Natural Language_" by Minervini et. al, and previous work trying to add inductive biases to Transformers such as "_Does Entity Abstraction Help Generative Transformers Reason?_" by Gontier et. al.

**Questions:**

- see "weakness (2)": Could the method settle on an "almost correct" syntactic tree to save time? and what would the effect of that be on performance?

- What is the vocabulary size of the primitives considered? Did you try more complex sets of logical primitives? What do you think the effect would be on time and performance?

- Do you have any hints of how to start thinking about representing probabilistic semantics like mentioned in the Limitation section?

---

> ### Author Response · Authors · 2023-11-19
>
> We sincerely thank you for the constructive comments and the acknowledgment of the significance of our work. Below, we provide detailed replies to your comments and questions.
>
> > I would suggest the authors provide at least 1 other neuro-symbolic method to compare against to make the results more significant. The work from Minervini et. al. on Greedy Theorem Provers, or other variants could potentially be used as a baseline for some of these tasks.
>
> Thank you for the suggestion! We appreciate your concern and will check Minervini et. al.'s work on Greedy Theorem Provers and other neural-symbolic methods to add more neural-symbolic baselines.
>
> > The ambiguous explanation of how the search for a GSS tree is terminated.
>
> The search for a GSS tree stops when one of the following conditions is satisfied:
> 1. the found GSS can generate the given ground truth `y`, i.e., the root node's value is equal to `y`;
> 2. the search steps exceed the pre-defined maximum searching steps.
>
> > It would be beneficial to explain why the three modules of NSR exhibit equivariance and recursiveness.
>
> Equations 1, 2, and 3 demonstrate that in all three modules of the NSR system, the joint distribution is factorized into a product of several independent terms. This factorization process makes the modules naturally adhere to the principles of equivariance and recursiveness, as outlined in Definitions 3.1 and 3.2.
>
> > Suggestions: In Table 3, for the task of compositional translation, it would be interesting to also evaluate the performance of a vanilla transformer like in the previous tables. Discuss previous neuro-symbolic works such as Neural Theorem Provers (NTPs) and Greedy NTPs.
>
> Thank you for the suggestions! We will incorporate them in the revision.
>
> > What is the vocabulary size of the primitives considered? Did you try more complex sets of logical primitives? What do you think the effect would be on time and performance?
>
> The total number of primitives is 15. Whether adding more primitives is beneficial or detrimental depends on their relevance to the task at hand. Including primitives that aren't relevant can lead to longer search time, or could even cause the search to fail if it exceeds the maximum number of search steps allowed. On the other hand, if the new primitives are relevant and useful, they can shorten the length of the program and accelerate the search process.
>
> > Do you have any hints of how to start thinking about representing probabilistic semantics like mentioned in the Limitation section?
>
> Probabilistic programming languages (PPL) or frameworks might be useful for representing probabilistic semantics. PPLs are designed to handle uncertainty in a structured and systematic manner. They integrate probabilistic models with traditional programming concepts, enabling users to represent and manipulate uncertainty explicitly in their models.

---

> > ### Comment · Reviewer_WJqb · 2023-11-22
> > **Thanks for the clarifications**
> >
> > Thank you for taking the time to address my questions and clarify some aspects. It is more clear now. Looking forward to the comparison with other neuro-symbolic baselines (and vanilla transformers for table 3)

---

> > > ### Author Response · Authors · 2023-11-23
> > >
> > > We have updated the performance of the vanilla Transformer to Table 3 in the paper revision: Transformer is slightly better than Seq2Seq (LSTM, GRU).
> > >
> > > For other neuro-symbolic baselines, we have found that it is hard to directly adapt NTPs or Greedy NTPs to the considered datasets, since NTPs and Greedy NTPs are designed for proving queries to knowledge bases. Instead, we are investigating the more flexible Neural Module Networks [1,2] to build other neuro-symbolic baselines. We will update them in the paper revision once we get reasonable results for these baselines.
> > >
> > > [1] Andreas et al. "Neural module networks." CVPR 2016.
> > >
> > > [2] Johnson et al. "Inferring and executing programs for visual reasoning." ICCV 2017.

---

### Official Review · Reviewer_Gi7u · 2023-10-29

**Soundness:** 1 poor
**Presentation:** 1 poor
**Contribution:** 2 fair
**Rating:** 3
**Confidence:** 3

**Summary:**

This paper presents a neuro-symbolic architecture called the NSR which consists of 3 steps:
1. perception module to convert raw input into symbols
2. a parser to compute a syntax tree over symbols
3. a program induction module to convert this syntax over induced symbols into a program which can then convert an input into an output deterministically.

Each of these components are separate probabilistic modules (though details about what these models are exactly is unclear from the paper). From results on 3 tasks, we see improvements on generalization compared to neural models.

**Strengths:**

The subject matter of the paper is to make progress towards improving compositional generalization in learnt models, which is a very important area.

**Weaknesses:**

*Presentation is unclear*: There are very few details about the actual approach in the Section-3 (and Figure-1) to fully understand what exactly the model is (See questions). Unfortunately, because there is a lack of details around the approach, it is hard to do a thorough assessment of this work, and I request the authors to revise their draft.

Moreover, the paper spends too much time (and math notation) on simple definitions such as “equivariance” and “recursiveness” and on flagposting “hypothesis” statements. Not necessarily cause for rejection, but I highly suggest that these be moved into an appendix, so more time is spent on explaining the approach.



*How general is this approach*: Most of the experiments here are on datasets where symbolic approaches are likely to help, but it is unclear how well this approach would do for natural language semantic parsing tasks such as GeoQuery. I'm not fully opposed to having experiments that are only on these programmatic datasets, but it would be good to have an extended discussion on what the symbols and programs look like for more natural data distributions.

**Questions:**

Here are some details I could not get from Section-3:

- What exactly are the symbols in T for each of the datasets?
- Is every raw input mapped to a single symbol or is there a consolidation step where multiple raw inputs can be associated with the same symbol?
- What models are used to parameterize all of the distributions in Eq~4? Are these neural networks?
- What is the overall parameter count?
- How does inference work for this model?
- How does this compare to other neuro-symbolic systems, for example "Neuro-Symbolic Concept Learner" from Mao et al. 2019?

---

> ### Author Response · Authors · 2023-11-19
> **[1/2] Clarifications for the questions on the proposed approach**
>
> We sincerely thank you for reviewing our paper and providing constructive feedback. Below, we provide detailed replies to your comments and hope these replies can resolve your major concerns.
>
> > *Presentation is unclear*: few details about the actual approach in Section-3 (and Figure-1), highly suggest moving simple definitions and statements to the Appendix and spending more space on explaining the approach.
>
> Thank you for your feedback! We appreciate your suggestions and will incorporate them into the revision. Below, we provide detailed clarifications to your questions regarding our proposed approach.
>
> - > What exactly are the symbols in T for each of the datasets?
>
> Figure-1 shows the examples of `T  = <(x,s,v), e>` for each dataset.
>
> For SCAN, `x` is the input word, e.g., `left`; `s` is the symbol id, e.g., `5`; `v` is the output actions, e.g., `[LTURN,JUMP]`. `(x,s,v)` forms a node, e.g., `left 5 [LTURN,JUMP]` in the first example of Figure-1. The edge `e` is the arrow, e.g., `left 5 [LTURN,JUMP] -> jump 4 [JUMP]`, which means the value of the node `jump 4 [JUMP]`, i.e., `[JUMP]`, is an input to the node `left 5 [LTURN,JUMP]`.
>
> For PCFG, `x` is the input word, e.g., `swap`; `s` is the symbol id, e.g., `51`; `v` is the output letters, e.g., `[C,B,A]`. `(x,s,v)` forms a node `swap 51 [C,B,A]` in the second example of Figure-1. The edge `e` is the arrow, e.g., `swap 51 [C,B,A] -> A 0 [A]`, which means the value of the node `A 0 [A]`, i.e., `[A]`, is an input to the node `swap 51 [C,B,A]`.
>
> For HINT, `x` is the input image, e.g., the image of `*`; `s` is the symbol id, e.g., `12`; `v` is the output number, e.g., `27`. `(x,s,v)` forms a node `* 12 27` in the third example of Figure 1. The edge `e` is the arrow, e.g., `* 12 27 -> 3 3 3`, which means the value of the node `3 3 3`, i.e., `3`, is an input to the node `* 12 27`.
>
> - > Is every raw input mapped to a single symbol or is there a consolidation step where multiple raw inputs can be associated with the same symbol?
>
> Currently, every raw input is mapped to a single symbol.
>
> - > What models are used to parameterize all of the distributions in Eq~4? Are these neural networks?
>
> \theta_p and \theta_s are parameterized by neural networks. \theta_l are functional programs, as illustrated by Figure 4(b).
>
> - > What is the overall parameter count?
>
> For SCAN and PCFG, the parameter count is ~0.2M, mainly from the dependency parser. For HINT, the parameter count is ~11M, mainly from the image encoder ResNet-18.
>
> - > How does inference work for this model?
>
> As illustrated in Figure 2, the neural perception module first maps the input x, e.g., a handwritten expression in Figure 1 (3) HINT, to a symbol sequence, `2 + 3 * 9`. The dependency parsing module then parses the symbol sequence into a tree in the form of dependencies, e.g., `+ -> 2 *`, `* -> 3 9`. Finally, the program induction module uses the learned programs for each symbol to calculate the values of the nodes in the tree in a bottom-up manner, e.g., `3 x 9 => 27, 2 + 27 => 29`.
>
> - > How does this compare to other neuro-symbolic systems, for example, "Neuro-Symbolic Concept Learner" from Mao et al. 2019?
>
> Mao et al. 2019's NS-CL model relies on a pre-established domain-specific language (DSL) tailored for CLEVR, which limits its flexibility, as the meanings of operations like Filter, Relate, and Query are predefined and unchangeable. Moreover, NS-CL uses GRU for question parsing, a model that prior research [1,2] has identified as less effective in generalizing to longer sequences. In contrast, our model operates with minimal domain-specific knowledge, learns the semantics of symbols directly from data, and demonstrates significantly improved generalization capabilities.
>
> [1] Lake, et al. Generalization without systematicity: On the compositional skills of sequence-to-sequence recurrent networks. ICML (2018).
>
> [2] Li, et al. A minimalist dataset for systematic generalization of perception, syntax, and semantics. ICLR (2023).

---

> ### Author Response · Authors · 2023-11-19
> **[2/2] An example of how our model might work for natural language semantic parsing tasks such as GeoQuery**
>
> > *How general is this approach*: it would be good to have an extended discussion on what the symbols and programs look like for more natural data distributions, like GeoQuery.
>
> To discuss how our approach can generalize to natural language semantic parsing tasks, we construct the following example of how our model might work for GeoQuery.
>
> Question: `How many rivers are in California?`
>
> SQL query: `SELECT count(*) FROM rivers WHERE state = 'California'`
>
> Here's how our framework might model this example:
> 1. The Perception module simplifies the question by removing unnecessary grammatical details:
> ```
> how-many rivers be in California
> ```
>
> 2. The Parsing module breaks down the simplified sentence into a parse tree.
> 3. The Semantic module then generates the query step-by-step in a bottom-up manner. Here's the parse tree with intermediate results:
> ```
> (be => [SELECT count(*) FROM rivers WHERE state = 'California']
>     (rivers => [SELECT count(*) FROM rivers]
>         how-many => [SELECT count(*)]
>     )
>     (in => [state = 'California']
>         California => ['California']
>     )
> )
> ```
>
> As discussed in Limitations of Section 5, training the entire framework from scratch on natural language datasets can be challenging. However, we have observed that the parse tree is quite similar to standard dependency structures. As a result, we can leverage existing parsers to initialize the modules in our framework, which could accelerate the training process.
>
> We hope the above clarifications address your concerns and thank you again for your valuable suggestions. Please let us know if there is any further question.

---

> > ### Author Response · Authors · 2023-11-22
> > **Do our responses address your concerns? Thank you again for your valuable feedback!**
> >
> > Dear Reviewer Gi7u,
> >
> > As we are approaching the end of the author-reviewer discussion phase, we want to check out if our responses address your concerns well. Are there any further clarifications we could provide? We look forward to your feedback and please feel free to let us know if we have missed any issues.
> >
> > Thank you again for your valuable feedback!
> >
> > Best,
> >
> > Authors

---

### Official Review · Reviewer_brEA · 2023-10-31

**Soundness:** 4 excellent
**Presentation:** 2 fair
**Contribution:** 3 good
**Rating:** 8
**Confidence:** 2

**Summary:**

This paper describes a new neurosymbolic model, NSR, which consists of (1) a task-dependent model mapping from inputs to strings; (2) the Chen-Manning dependency parser; (3) a program induction module that is somehow based on DreamCoder. This pipeline is trained by gradient-based optimization (SGD?) using Metropolis-Hastings sampling to estimate the gradient. The proposed method performs well across four tasks, SCAN, PCFG, and HINT, and an artificial machine translation task.

**Strengths:**

This is a very interesting approach that achieves very good results on the four tasks tested. In every setting, their model either does the best, or is tied with NeSS because both models achieve 100% accuracy.

**Weaknesses:**

Many statements are made like, "This stark discrepancy underscores the pivotal role and efficacy of symbolic components—specifically, the symbolic stack machine in NeSS and the GSS in NSR—in fostering systematic generalization." But, for an outsider, no explanation is given for why the symbolic components actually lead to better generalization. I would like to see some more explanation or analysis to back this up.

The program induction module is not described in detail; in equation (3), what is the p in the right-hand side? When you say that you "leverage" DreamCoder, do you mean that this module simply is DreamCoder?

Figure 3: image is wrong?

**Questions:**

Section 3.3: How do you use the gradients? Is this SGD?

Why is abduction called that? It seems different from abductive reasoning?

Def 3.2: Isn't "compositionality" a more usual word for this? "Recursive" has a totally different meaning in theory of computation.

---

> ### Author Response · Authors · 2023-11-19
>
> We are grateful for your review of our paper and your constructive feedback. Your questions are answered in detail below:
>
> >More explanation or analysis on why the symbolic components actually lead to better generalization.
>
> We think the effectiveness of symbolic components in fostering systematic generalization in neural networks, such as the symbolic stack machine in NeSS and the Grounded Symbolic System (GSS) in NSR, stems from their ability to introduce *structured, rule-based processing* into otherwise fluid neural computations. These symbolic systems act as a framework within which neural networks can operate more predictably and consistently, thereby enhancing their generalization capabilities. Besides, symbolic components can mitigate overfitting by imposing rule-based constraints on the learning process. These constraints ensure that the model does not merely memorize the training data but learns the underlying rules and structures, thereby improving its ability to generalize to new, unseen data.
>
> > In equation (3), what is the p in the right-hand side?
>
> `p` denotes the probability.
>
> > When you say that you "leverage" DreamCoder, do you mean that this module simply is DreamCoder?
>
> The original DreamCoder does not support noisy examples, so we modified its search process to tolerate a ratio of noisy examples.
>
> We kindly refer the reviewer to Appendix Section A for more details on the program induction module.
>
> > Figure 3: image is wrong?
>
> Yes, thank you for pointing it out! We will correct it in the revision.
>
> > Section 3.3: How do you use the gradients? Is this SGD?
>
> Yes, we use the Adam optimizer in practice.
>
> > Why is abduction called that?
>
> Naturally, this is a process of abductive reasoning. Consider the input `x` and output `y` as observations. The goal is to identify the `T` that most likely explains these observed values of `x` and `y`. This suggests that the discovered `T` may not necessarily match the actual ground-truth value of `T`.
>
> > Def 3.2: Isn't "compositionality" a more usual word for this?
>
> Thank you for pointing it out! The word "compositionality" is indeed a more appropriate choice for Def 3.2 and we will update it in the revision.

---

### Official Review · Reviewer_ih16 · 2023-11-01

**Soundness:** 3 good
**Presentation:** 4 excellent
**Contribution:** 2 fair
**Rating:** 6
**Confidence:** 4

**Summary:**

This paper introduces the Neural-Symbolic Recursive Machine (NSR), a model for systematic generalization in sequence-to-sequence tasks. The key innovation is representing the problem as a Grounded Symbol System (GSS) with combinatorial syntax and semantics that emerge from training data. The NSR incorporates neural modules for perception, parsing, and reasoning that are jointly trained via a deduction-abduction algorithm. Through architectural biases like recursiveness and equivariance, the NSR achieves strong systematic generalization on tasks including semantic parsing, string manipulation, arithmetic reasoning, and compositional machine translation.

Overall, the paper presents a novel neural-symbolic architecture that combines beneficial inductive biases from both neural networks and symbolic systems to achieve human-like generalization and transfer learning abilities. The experiments demonstrate strengths on challenging benchmarks designed to test systematic generalization.

**Strengths:**

- Compositional generalization is an interesting and important direction to explore, which should be one of the most important capabilities of human. Therefore, the problem and the research direction is important.
- The Neural-Symbolic Recursive Machine (NSR) model is a novel model architecture centered around representing problems as grounded symbol systems. The deduction-abduction training procedure for coordinating the modules is an original contribution for jointly learning representations and programs.
- The paper clearly explains the limitations of existing methods, the need for systematic generalization, and how the NSR model aims to address this. The model description and learning algorithm are well-explained. The experiments and analyses effectively demonstrate the claims.
- The paper is technically strong, with rigorous definitions and detailed exposition of the model components and learning algorithm.
- The experiments systematically test generalization across four datasets with carefully designed splits. The analyses provide insights into when and why the NSR architecture generalizes better than baselines.

Overall, this is a technically strong and well-written paper that makes both conceptual and practical contributions towards an important research direction.

**Weaknesses:**

Although I understand that compositional generalization is currently driven primarily by synthetic datasets like SCAN, I would still like to see the application of this method in real-world scenarios. For example, could it achieve significantly better generalization performance compared to conventional seq2seq models on real machine translation tasks?

**Questions:**

N/A

---

> ### Author Response · Authors · 2023-11-19
> **Thank you for reviewing our paper!**
>
> We sincerely thank you for reviewing our paper and acknowledging the novelty and effectiveness of the proposed method. Below, we provide a detailed response to your question.
>
> > Could it achieve better generalization performance than conventional seq2seq models on real machine translation tasks?
>
> Our proposed model achieves much better performance on a compositional machine translation task, as described in Section 4.4. Table 3 shows the comparison: our model can achieve perfect generalization accuracy (100\%), while the conventional seq2seq model performs badly (12\%).

---

> > ### Comment · Reviewer_ih16 · 2023-11-22
> > **Official Comment from Reviewer ih16**
> >
> > Thanks the author for response! Yes I understand it is a more challenging task than SCAN, but it is still a synthetic benchmark. For a reference, maybe some datasets achieving the complexity of CFQ [1] in machine translation would be a great fulfillment. But I understand this may be beyond the scope of this article. Overall, the quality of this paper is good and should be accepted.
> >
> > [1]. Measuring Compositional Generalization: A Comprehensive Method on Realistic Data, ICLR 2022

---

> > > ### Author Response · Authors · 2023-11-22
> > >
> > > Thank you for the reference to the CFQ dataset. While our current work indeed focuses on a synthetic benchmark, which is crucial for controlled experimentation and incremental progress, incorporating datasets like CFQ could indeed provide a more comprehensive view of a model's capabilities in handling complex, real-world scenarios.
> > >
> > > The positive assessment of our paper is encouraging. We believe that even with its current focus, the research contributes meaningfully to the field, and we're glad to hear that you see its merit for acceptance. We look forward to exploring more complex datasets and challenges in our future work, building on the insights gained from this study.

---

### Author Response · Authors · 2023-11-23
**A summary of paper revision**

We thank all reviewers for their insightful comments and acknowledgment of our contributions. We've collected the suggestions until now and revised our manuscript accordingly (highlighted in red in the uploaded revision pdf). Detailed responses to each reviewer's concerns are carefully addressed point-by-point. Below summarize the major updates we've made:

- Discussion and citations to more related works: (**Related Work**)
   > - [A] Rocktasche et al. *End-to-end differentiable proving.* NeurIPS 2017. (NTPs)
   > - [B] Minervini et al. *Differentiable reasoning on large knowledge bases and natural language.* AAAI 2020. (Greedy NTPs)
   > - [C] Mao et al. *The neuro-symbolic concept learner: Interpreting scenes, words, and sentences from natural supervision.* ICLR 2018. (NS-CL)
   > - [D] Gontier et al. *Does entity abstraction help generative transformers reason?* TMLR 2022.

- Update Figure 3 to use the correct image. (**Figure 3**)

- Replace "recursiveness" with "compositionality". (**Definition 3.2**)

- Move simple definitions to the Appendix and add more details on the actual approach, including: (**Section 3**)
  > - The explanation of the symbols in `T = <(x,s,v), e>` for each dataset. (**Section 3.1**)
  > - Model inference process. (**Section 3.2**)
  > - The stop criteria of the search for a GSS tree. (**Section 3.3**)

 - The explanation of why the three modules of NSR exhibit equivariance and recursiveness. (**Appendix C**)

-  The performance of the vanilla Transformer for the compositional machine translation. (**Table 3**)

We believe that these changes have enhanced the quality of our manuscript and made it more accessible to the audience. Thank you again for your valuable feedback, and we look forward to any further guidance or suggestions you may have.

Best,

Authors

---

### Meta-Review · Area_Chair_MHU2 · 2023-12-06

**Metareview:**

The paper proposes a neural-symbolic architecture called the Neural-Symbolic Recursive Machine, consisting of a number of modules including perception, parsing, and reasoning. Experimental results show improved compositional generalization on a number of benchmarks. Weaknesses include that the clarity of the presentation can be improved further, and that the architecture might not be general enough, given that it is primarily evaluated on synthetic or domain-specific datasets. However, the paper's contribution in terms of novelty and experimental results is still strong enough for acceptance.

**Justification For Why Not Higher Score:**

The proposed architecture and experimental evaluation are not general enough to justify spotlight.

**Justification For Why Not Lower Score:**

The paper proposes a novel architecture and the experimental results show improved compositional generalization across multiple benchmarks.

---

### Decision · Program_Chairs · 2024-01-16

Accept (poster)